# All Things Considered: Detecting Partisan Events from News Media with Cross-Article Comparison

**Yujian Liu**[1]    **Xinliang Frederick Zhang**[2]    **Kaijian Zou**[2]
**Ruihong Huang**[3]    **Nick Beauchamp**[4]    **Lu Wang**[2]

[1]Computer Science, UC Santa Barbara, Santa Barbara, CA
[2]Computer Science and Engineering, University of Michigan, Ann Arbor, MI
[3]Computer Science and Engineering, Texas A&M University, College Station, TX
[4]Department of Political Science, Northeastern University, Boston, MA
[1]yujianliu@ucsb.edu, [2]{xlfzhang,zkjzou,wangluxy}@umich.edu
[3]huangrh@tamu.edu, [4]n.beauchamp@northeastern.edu

## Abstract

Public opinion is shaped by the information news media provide, and that information in turn may be shaped by the ideological preferences of media outlets. But while much attention has been devoted to media bias via overt ideological language or topic selection, a more unobtrusive way in which the media shape opinion is via the strategic inclusion or omission of *partisan events* that may *support* one side or the other. We develop a latent variable-based framework to predict the ideology of news articles by comparing multiple articles on the same story and identifying partisan events whose inclusion or omission reveals ideology. Our experiments first validate the existence of partisan event selection, and then show that article alignment and cross-document comparison detect partisan events and article ideology better than competitive baselines. Our results reveal the high-level form of media bias, which is present even among mainstream media with strong norms of objectivity and nonpartisanship. Our codebase and dataset are available at https://github.com/launchnlp/ATC.

## 1 Introduction

News media play a critical role in society not merely by supplying information, but also by selecting and shaping the content they report (de Vreese, 2004; DellaVigna and Kaplan, 2007; DellaVigna and Gentzkow, 2009; Perse and Lambe, 2016). To understand how media bias affects media consumers (Gentzkow and Shapiro, 2006; Gentzkow et al., 2015), we must understand not just how media ideology affects the presentation of news stories on a surface level, such as the usage of partisan phrases or opinions, but also the less obvious process of content selection (Fan et al., 2019; Enke, 2020). Content selection, such as what events that are related to the main story and should be included

---

**News Story:** *Biden pushes for gun legislation after visiting Uvalde.*

**The Washington Post** (left):
E1: [Jaydien]$_{ARG0}$, ..., [said]$_{pred}$ [he asked the president: "Could you please make our schools safer and send more police, please?"]$_{ARG1}$
E2: [Biden]$_{ARG0}$ ... [noting]$_{pred}$: "[You couldn't buy a cannon when the Second Amendment was passed]$_{ARG1}$."

**New York Post** (right):
E1: [You]$_{ARG0}$ couldn't [buy]$_{pred}$ [a cannon]$_{ARG1}$ when the Second Amendment was passed.
E2: Biden has made that claim before, ..., and they have been repeatedly [declared]$_{pred}$ [false]$_{ARG1}$ [by fact-checkers]$_{ARG0}$.

Figure 1: Article snippets by different media on the same story. Events are represented by triplets of ⟨ARG0, predicate, ARG1⟩. Events favoring left and right sides are highlighted in blue and red. Events in black are reported by both media and not considered as partisan.

---

in the report, has recently become a focus of study in political science. Numerous studies point out that media selectively report information that is flattering to a particular political party or ideology, which may consequently shift audience beliefs and attitudes (Broockman and Kalla, 2022; Baum and Groeling, 2008; Grossman et al., 2022; D'Alessio and Allen, 2006). However, most existing work either requires manual inspection of reported content (Broockman and Kalla, 2022), or relies on simple tools for coarse analyses, such as overall slant and topic emphasis (Baum and Groeling, 2008; Grossman et al., 2022). As a result, these studies are either limited to a short time period, or are unable to provide a detailed understanding of content selection bias. Thus there remains a strong need for automatic tools that can analyze and detect how more complex content is selectively reported.

Rather than focusing on more superficial biases such as word, topic, or entity selection, we investigate here how media ideology affects their selection of which **events** to include for news reporting.

Events are the fundamental high-level components of the storytelling process (Prince, 2012), and their inclusion or omission shapes how a news story is perceived. In line with previous analysis of partisan selection bias in the literature (Broockman and Kalla, 2022), we define **partisan events** as *selectively reported events that are favorable to a media organization's co-partisans or unfavorable to counter-partisans*. When there are many potentially relevant events, which subset are included in an article fundamentally affects how readers interpret the story, and can reveal a media outlet's stance on that topic and their ideology (Mullainathan and Shleifer, 2005; McCombs and Reynolds, 2008; Entman, 2007). One example of event-selection bias is shown in Fig. 1, where a Washington Post article includes a survivor's request to impose gun control (*pro*-gun control), whereas a New York Post article claims Biden's statement as false (*pro*-gun rights).

This paper has two major goals: (1) examining the *relation between event selection and media ideology*, and (2) formulating a task for *partisan event detection in news articles* and developing computational methods to automate the process. For the first goal, we verify the existence of partisan event selection by measuring how event selection affects the performance of media ideology prediction. Specifically, we represent articles using triplets of ⟨ARG0, predicate, ARG1⟩, denoting the set of events they report with participating entities (e.g., in Fig. 1). This representation is shown to be effective in narrative understanding (Chambers and Jurafsky, 2008; Mostafazadeh et al., 2016). We conduct *two studies*. First, we compare article-level ideology prediction performance by using events within a single article vs. contextualizing them with events in other news articles on the same story but reported by media with different ideologies, inspired by the observation that biased content should be evaluated against other media (Larcinese et al., 2011). We show that the latter setup yields significantly higher F1 scores, suggesting that *cross-article comparison* can identify partisan events and thereby produce more accurate ideology prediction. Second, we *annotate an evaluation* dataset of 50 articles that focus on two recent political issues, where in total we manually label 828 *partisan events* out of 1867 sentences from all articles. *Testing* on this dataset, we show that removing partisan events from the articles hurts ideology prediction performance significantly more than removing similar amounts of randomly selected events.

For the second goal, the most critical challenge in developing computational tools to identify partisan events is the *lack of annotation*, where manually labeling a large-scale dataset requires domain expertise and is highly time-consuming. For that reason, we use **latent variables** to represent whether an event is partisan or not, and propose to jointly infer partisan events and predict an article's ideology. Our models are trained using article-level ideology labels only, which are easier to obtain, and they *do not require any human annotation of partisan events*. We compare two approaches (Chen et al., 2018; Yu et al., 2019) to train latent variable models and explore two methods for further improvement: (1) steering the model toward events that are selected only by one side, which are more likely to be partisan, and (2) providing prior ideology knowledge with pretrained event representations.

We conduct experiments on two existing news article datasets (Liu et al., 2022; Fan et al., 2019) and our newly annotated data with partisan events (*test only*). Results indicate that latent variable models outperform all competitive baselines on both partisan event detection and ideology prediction, where cross-article event comparison is shown to be critical for both tasks. Analysis of the extracted partisan events reveals key challenges in detecting implicit nuanced sentiments and discerning event relations (e.g., main vs. background events), suggesting future research directions.

To the best of our knowledge, this is the first time that computational methods are developed for studying media bias at the event selection level. It is also the first time that automatic models are investigated to detect partisan events. Our results provide new insights into a high-level form of media bias that may be present even in apparently nonpartisan news, enabling a new understanding of how news media content is produced and shaped.

## 2 Related Work

**Media Bias Understanding and Detection.** Research in political science, economics, and communication has extensively demonstrated the relationship between news media and ideological bias (Mullainathan and Shleifer, 2005; Gentzkow et al., 2014). According to Broockman and Kalla (2022), there are three common strategies news media use to affect readers: *Agenda setting* (Mc-

Combs and Shaw, 1972) refers to when the public's perception of a topic's overall significance is shaped by the amount of news coverage spent on that topic (Field et al., 2018; Grimmer, 2010; Quinn et al., 2010; Kim et al., 2014). *Framing* concerns how media highlight some aspects of the same reality to make them more salient to the public (Entman, 1993; Tsur et al., 2015; Baumer et al., 2015; Card et al., 2015; Liu et al., 2019a). Finally, *partisan coverage filtering* is used by media to selectively report content that is flattering to their co-partisans or unflattering to opponents. While there is a certain amount of conceptual overlap among these three categories, this work focuses primarily on the third: the selection of which events relevant to the main stories to report, and how that reveals a media outlet's ideology and stance. Compared to previous work in agenda setting, which mainly focuses on the topics of news articles (Field et al., 2018; Kim et al., 2014), our partisan event study focuses on a more thoughtful process for information filtering. Event selection is also subtler than framing, since framing examines how a perspective is evoked through particular phrases (Card et al., 2015), whereas partisan event detection requires both event extraction and cross-article comparison.

While partisan coverage filtering has been studied in political science, detecting it requires human efforts to review all news content (Broockman and Kalla, 2022; Baum and Groeling, 2008), making these methods unscalable and only applicable to short time periods. Grossman et al. (2022) automate the process, but use predefined lists of phrases and simple topic models to determine the overall slant and topic of a news report, which cannot capture more tactful content selection like events. Most recently, a contemporaneous work (Zou et al., 2023) also explores the partisan events within news articles, but they mainly curate a larger-scaled annotated dataset to support fine-tuning models on the labeled events. Compared to these works, we operate with *more nuanced factual details* than phrases and topics, and we treat partisan events as latent variables and automatically detect them from news articles with methods that are scalable to large quantities of news.

Another line of work that is similar to ours is the detection of *informational bias* (Fan et al., 2019; van den Berg and Markert, 2020), defined as "tangential, speculative, or background information that sways readers' opinions" (Fan et al., 2019).

Our work differs in two important aspects: First, their "informational bias" can occur in any text span, and detecting speculative information often requires complex inference and also depends on specific wording. By contrast, by focusing on the presence or absence of events, we target concrete units of potentially partisan information, which can be more easily validated and understood by readers. Second, they train supervised models on annotated biased content, while our latent variable models do not need any labels on partisan events.

**Ideology Prediction with Text.** Many computational models have been developed to predict ideology using textual data (Gentzkow and Shapiro, 2010; Gerrish and Blei, 2011; Ahmed and Xing, 2010; Nguyen et al., 2013). Recent work, for instance, leverages neural networks to incorporate phrase-level ideology (Iyyer et al., 2014), external knowledge from social media (Kulkarni et al., 2018; Li and Goldwasser, 2019), and large-scale language model pretraining (Liu et al., 2022; Baly et al., 2020). However, most of this computational work focuses directly on ideology prediction, with little attention to the higher-level processes underlying media bias. In particular, ideology prediction may fail for many mainstream media outlets who eschew overtly ideological language, and instead may bias readers only via a more sophisticated information selection procedure at the event level. We demonstrate that incorporating story-level context enables global content comparison over political spectrum, and benefits both partisan event detection and ideology prediction.

## 3 Event Selection Effect Study

In this section, we verify the *existence of partisan event selection* by examining partisan events' influence on ideology prediction. Using events extracted from articles, we design a model that predicts ideology with single- or multi-article context (§3.1), based on the assumption that comparing events included by different media may reveal their ideological leanings. We then manually annotate a dataset with partisan events in news stories (§3.2). Using this dataset and two existing corpora, we show that cross-article content comparison can reveal potential partisan events and removing partisan events hurts ideology prediction (§3.3).

## 3.1 Ideology Prediction with Events

We build on the narrative embedding model in Wilner et al. (2021) and extend it to include story level context by adding article segment, event frequency, and event position embeddings. This allows us to gauge the effect of partisan events' presence or absence on ideology prediction.

**Event Extraction.** We follow prior work (Zhang et al., 2021) to train event extractor on the MATRES dataset (Ning et al., 2018). Our extractor achieves an F1 score of 89.53, which is on par with the state-of-the-art performance (90.5) (Zhang et al., 2021). See details in Appendix B.1.

**Ideology Prediction.** Given $N$ articles $a_1, \ldots, a_N$ that report on the same news story, we denote events in article $a_i$ as $x_1^{(i)}, \ldots, x_{L_i}^{(i)}$, where $L_i$ is the number of events in article $a_i$. We first use a DistilRoBERTa model (Sanh et al., 2019) to get the embedding $\mathbf{e}$ for an event.[1] Concretely, we input the sentence that contains the event to DistilRoBERTa and get the embeddings $\mathbf{e}_{pred}, \mathbf{e}_{arg0}, \mathbf{e}_{arg1}$ for predicate, ARG0, and ARG1 by taking the average of last-layer token embeddings. If a sentence has multiple events, we mask out other events' tokens when encoding one event, so that the information in one event does not leak to others. We then get $\mathbf{e} = \mathbf{W}[\mathbf{e}_{pred}; \mathbf{e}_{arg0}; \mathbf{e}_{arg1}]$, where ; means concatenation and $\mathbf{W}$ is learnable.[2] We then input all events in one article or all articles on the same story to another transformer encoder (Vaswani et al., 2017) to get contextualized $\mathbf{c}$ for each event:

$$[\mathbf{c}_1^{(1)}, \ldots, \mathbf{c}_{L_N}^{(N)}] = \texttt{Transformer}\left([\mathbf{e}_1^{(1)}, \ldots, \mathbf{e}_{L_N}^{(N)}] + E\right) \quad (1)$$

where `Transformer` is a standard transformer encoder trained from scratch (details in Appendix B.2) and $E$ contains three types of embeddings: **Article embeddings** distinguish the source by associating the index of the article with its events, with a maximum of three articles per story. **Frequency embeddings** highlight the prevalence of events by signaling if an event appears in only one article, more than one but not all articles, or all articles that report the same story. We train one embedding for each category and use lexical matching to determine common events. Finally, **position embeddings** represent the relative position of an event in the article, e.g., partisan events may appear later in the reports. All embeddings are learnable (de-

---

[1]We use DistilRoBERTa due to computational constraints.
[2]We use a zero vector if ARG0 or ARG1 does not exist.

| | AllSides | Basil | PEvent (ours) |
|---|---|---|---|
| # stories | 2,221 | 67 | 25 |
| # articles | 5,361 | 134 | 50 |
| # events detected per article | 66.82 | 48.71 | 60.70 |

Table 1: Statistics for AllSides training set, Basil (test only), and PartisanEvent (test only). AllSides **test** set contains 1,416 articles.

tails in Appendix B.2). Note that Eq. 1 describes the model with story level context as it includes all events in all articles. We also experiment with models that only use events in one article. Finally, the model predicts article's ideology using average representation of all events in the article.

## 3.2 Partisan Event Dataset Annotation

Since there is no dataset with partisan event annotations for news articles, we manually label a **Partisan Event** (**PEvent**) dataset with 50 articles (1867 sentences) covering two controversial events happened in the U.S. in 2022: a mass shooting in Texas, and the overturn of *Roe v. Wade*. Note that PEvent contains articles from a separate and **later** time than the training data with *ideology prediction* objective. PEvent is only used for **evaluation purposes** on the task of *partisan event detection*.

Since labeling partisan events is costly, which requires both domain knowledge and news annotation experience, we only focus on two broad high-profile topics where the partisanship of all constituent events is already known to coders experienced with US politics. We acknowledge that a dataset with diverse topics would be useful, but will leave this for the future work. We collect articles from AllSides Headline Roundups section,[3] where groups of three articles that report the same news story are carefully selected by editors to demonstrate "how opposite sides of the media are discussing or framing a subject". For each story, we discard the center ideology article due to a lack of consensus of what constitutes center ideology by the community. The remaining two articles, together with extracted events, are provided to two college students who have prior news article annotation experience and have gone through careful training of the annotation tasks. They are instructed to first label article ideology,[4] and then partisan

---

[3]https://www.allsides.com/blog/how-does-allsides-create-balanced-news.
[4]We intentionally annotate articles' ideology rather than using media-level ideology to ensure accurate ideology labels.

| | AllSides | Basil | PEvent |
|---|---|---|---|
| Single-article | $64.10 \pm 3.51$ | $55.08 \pm 6.01$ | $44.37 \pm 2.60$ |
| $+pos.$ | $64.37 \pm 0.75$ | $54.78 \pm 2.38$ | $45.77 \pm 3.46$ |
| Multi-article | $79.52 \pm 1.52$ | $64.91 \pm 1.78$ | $76.64 \pm 3.16$ |
| $+art.$ | $\underline{88.61} \pm 0.84$ | $67.30 \pm 2.45$ | $\mathbf{85.19} \pm 2.28$ |
| $+art. + fre.$ | $\mathbf{88.64} \pm 0.56$ | $\underline{68.05} \pm 1.33$ | $\underline{83.60} \pm 1.67$ |
| $+art. + fre. + pos.$ | $88.49 \pm 0.74$ | $\mathbf{68.50} \pm 2.07$ | $83.59 \pm 1.67$ |

Table 2: Macro F1 scores for article ideology prediction (average of 5 runs). **Best** results are in bold and second best are underlined. $art.$, $fre.$, and $pos.$ refer to article, frequency, and position embeddings in §3.1.

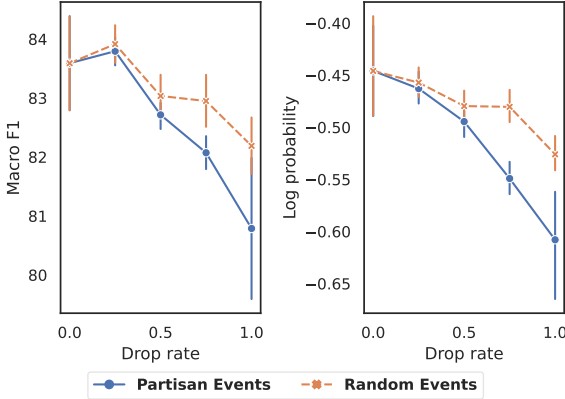

Figure 2: Performance (10 runs) after removing the same number of partisan events and random events. Performance drops for both settings, but removing partisan events leads to more severe performance regression.

events. During annotation, we only annotate left partisan events for left articles and vice versa. Finally, a third annotator compares the annotations and resolves conflicts. Appendix C contains the full annotation guideline.

In total, **828 partisan events** are annotated out of 3035 events detected by our tool from 1867 sentences. Inter-annotator agreement calculated using Cohen's $\kappa$ (Cohen, 1960) is $0.83$ for article-level ideology. For partisan event labeling, two annotators achieve $\kappa = 0.43$, which is substantial agreement. After discussing with the annotators, we find that disagreement often occurs when one annotator is insufficiently confident and thus ends up labeling an event as non-partisan. Therefore during the disagreement resolution stage, an event is frequently deemed partisan if it is labeled by at least one annotator. This again highlights the subtlety of partisan event usage by media. On average, $16.56$ ($27.28\%$) events are annotated as partisan events per article. Among all partisan events reported by left-leaning media, $98.41\%$ are chosen only by the left side, and $95.09\%$ for the right media. We further check where partisan events are included in the articles, and find that they occur more frequently in the later parts of articles written by right-leaning media (displayed in Fig. 4 in the Appendix). *These findings validate our design in §3.1.*

### 3.3 Results for Ideology Prediction

We first compare ideology prediction performance using different model variants in §3.1 and then pick two to study the effect of removing partisan events.

**Effects of Cross-Article Event Comparison.** We train models on AllSides dataset collected in Liu et al. (2022), where media outlets' ideology is used as articles' ideology. We use articles before 2020 (inclusive) as training and dev data and articles after 2020 (exclusive) as test data. We also evaluate models on Basil (Fan et al., 2019), where ideology

is manually annotated similar to §3.2. Likewise, we remove articles of the center ideology. Table 8 presents the statistics for datasets used in this study.

We experiment with multi- and single-article variants of the model, depending on whether the transformer in Eq. 1 has access to events in all or one article. As shown in Table 2, multi-article models that allow content comparison across articles written by different media significantly outperform single-article models, demonstrating the benefits of adding story-level context to reveal partisan events that improve ideology prediction. Among multi-article models, article embeddings lead to the largest gain since it supports cross-article comparison. For experiments in the rest of this paper, we add position embedding for single-article models and all three embeddings for multi-article models.

**Effects of Removing Partisan Events.** Next, we investigate how would removing partisan events affect model's prediction on ideology. Intuitively, when having access to fewer partisan events, the model will be less confident in predicting correct ideologies. Concretely, we run the multi-article model on PEvent. We drop $m\%$ of **partisan events**, where $m = 25, 50, 75, 100$. We also run the same model and remove the same number of events randomly (**random events**). We then measure the macro F1 and log probability of true classes.

As shown in Fig. 2, removing partisan events hurts the performance more compared to removing random events. Moreover, the more partisan events are removed, the larger the performance gap is, which confirms that models exploit the presence of partisan events to discern ideology.

## 4 Latent Variable Models for Partisan Event Detection

The general idea of our latent variable models for partisan event detection is that the detected partisan events should be indicative of article's ideology, the removal of which would lower models' prediction confidence, according to our study in §3. We adopt two methods that are originally developed to extract rationales of model predictions (§4.2) for our task and further improve them by adding constraints on the usage of common events and adding prior knowledge of event-level ideology (§4.3).

### 4.1 Task Overview

We assume our data comes in the form of $(a, y)$, where $y$ is the ideology for article $a$. We extract events $\mathbf{x} = (x_1, \ldots, x_L)$ from article $a$ where $L$ is the number of events in the article. We define a **latent** random variable $m_i \in \{0, 1\}$ for each event $x_i$, and $m_i = 1$ means $x_i$ is a partisan event. The ideology prediction task aims at predicting $y$ using $\mathbf{x}$. The partisan event detection task focuses on predicting partisan indicators $\mathbf{m} = (m_1, \ldots, m_L)$.

### 4.2 Latent Variable Models

**Two-Player Model.** We adopt methods in rationale extraction, where rationale is defined as part of inputs that justifies model's prediction (Lei et al., 2016). We use the formulation in Chen et al. (2018), which tackles the rationale (partisan events in our model) extraction task from an information-theoretic perspective. In details, suppose a positive number $k$ is given, the goal is to extract $k\%$ of events that have the highest mutual information with label $y$ and treat them as partisan events. In other words, our partisan indicator $\mathbf{m}$ satisfies $|\mathbf{m}| = k\% * L$. Since optimizing mutual information is intractable, Chen et al. (2018) provides a variational lower bound as the objective instead:

$$\max_{\mathcal{E}_\theta, q_\phi} \sum_{(\mathbf{x}, y) \in \mathcal{D}} \mathbb{E}_{\mathbf{m} \sim \mathcal{E}_\theta(\mathbf{x})} \left[ \log q_\phi(y \mid \mathbf{m} \odot \mathbf{x}) \right] \quad (2)$$

where $\mathcal{E}_\theta$ is an extractor that models the distribution of $\mathbf{m}$ given $\mathbf{x}$, $q_\phi$ is a predictor that predicts $y$ given partisan events, $\mathcal{D}$ is the training set, and $\odot$ is the element-wise multiplication.

We parameterize both $\mathcal{E}_\theta$ and $q_\phi$ using the same model as in §3.1. For the extractor, we first get the embedding $\mathbf{e}$ for all events and then pass it to the transformer encoder to get contextualized event representations. A linear layer converts these represen-

tations to logits, from which we sample $k\%$ of them following the subset sampling method in Xie and Ermon (2019)—a differentiable sampling method that allows us to train the whole system end-to-end. At inference time, we select the top $k\%$ of events with the largest logits by the extractor. For the predictor, we again get event embeddings $\mathbf{e}$, but we input $\mathbf{m} \odot \mathbf{e}$ to the transformer encoder so that it only sees the sampled subset of events.

**Three-Player Model.** Among all events in the article, some may have spurious correlation with the ideology. For instance, the event "a CNN reporter contribute to this article" can almost perfectly reveal article's ideology. To prevent models from focusing on these shortcuts, we further investigate the method in Yu et al. (2019). Concretely, they propose a three-player model where a third complement predictor $q_\pi^c$ predicts ideology using the complement of partisan events, i.e., $(\mathbf{1} - \mathbf{m}) \odot \mathbf{x}$. The goal for both predictors is to correctly predict the ideology, i.e., maximize $\log q_\phi(y \mid \mathbf{m} \odot \mathbf{x})$ and $\log q_\pi^c(y \mid (\mathbf{1} - \mathbf{m}) \odot \mathbf{x})$. The objective for the extractor is to select $k\%$ of events that can predict $y$ while the remaining events cannot as in Eq. 3:

$$\max_{\mathcal{E}_\theta} \sum_{(\mathbf{x}, y) \in \mathcal{D}} \mathbb{E}_{\mathbf{m} \sim \mathcal{E}_\theta(\mathbf{x})} \left[ \log q_\phi(y \mid \mathbf{m} \odot \mathbf{x}) \right. \\ \left. - \log q_\pi^c(y \mid (\mathbf{1} - \mathbf{m}) \odot \mathbf{x}) \right]. \quad (3)$$

Intuitively, the extractor and the complement predictor play an adversarial game, and Eq. 3 drives the extractor to identify partisan events as comprehensive as possible so that the complement predictor cannot perform well. In fact, Yu et al. (2019) uses an explicit objective to penalize $\sum_i m_i$ when it deviates from $k\% * L$, but we find this objective does not work well with Eq. 3, leading to an extractor that either selects all events as partisan events or detects nothing. We thus modify it with the subset sampling method (Xie and Ermon, 2019) again. At inference time, we use $q_\phi$ for ideology prediction.

Both two-player and three-player models can have the single- and multi-article variants, depending on whether the extractor and predictors can access all events in a story or just from a single article. Appendix D details the training process.

### 4.3 Improving Partisan Event Detection

**Restricting Models from Picking Common Events.** As shown in Fig. 1, common background events and main events should not be considered as partisan events. We therefore explicitly prohibit models from selecting these events. Precisely, we

use the same lexical matching method as in §3.1 to find common events in the story. During training, we add an auxiliary objective that minimizes the probability of the extractor to predict events that appear in both left and right articles as partisan events, thus driving models to prefer events reported by only one side. We only apply this constraint to multi-article models since it requires story-level context to locate common events.

**Pretraining to Add Event Ideology Priori.** Prior knowledge, especially the media's stance on controversial topics, plays an important role in partisan content detection. Given that the AllSides training set is relatively small, it is unlikely for the model to gain such knowledge on a broad range of topics. We therefore pretrain a model on BIGNEWSALIGN dataset in Liu et al. (2022) to acquire prior knowledge at the event level.

BIGNEWSALIGN is a dataset with 1 million political news stories, and each story contains about 4 articles that report the same main event. We extract events in these articles and train a DistilRoBERTa model as in §3.1 to predict the ideology of each event, where we use article's ideology as event's ideology. Note that this model takes each event as input and does not consider any context information. Intuitively, it counts the reporting frequency of each event: If an event is reported more by left media, it has a higher probability of being left and vise versa. We use this pretrained model for initialization in the extractor and the predictor.

# 5 Experiments

**Tasks and Datasets.** Similar to §3.3, we train all models solely on AllSides and evaluate on AllSides test set, Basil, and our partisan event dataset (PEvent). We measure ideology prediction performance on all three datasets and partisan event detection performance on PEvent.

**Evaluation Metrics.** For ideology prediction, we measure the macro F1 score at the article level. For partisan event detection, we measure the F1 score for the positive class, i.e., partisan event.

**Baselines.** We consider the following baselines: (1) We **random**ly predict partisan events with a 0.3 probability, and randomly predict ideology for the article. (2) **Event-prior** is the pretrained event model with ideology priori in §4.3. We run it to get the probability of each event being left and right. We then consider the $30\%$ of events with the most skewed distribution as partisan events. Finally, we

| | Ideology Prediction | | | Event |
|---|---|---|---|---|
| | AllSides | Basil | PEvent | PEvent |
| Random | $49.83_{\pm1.65}$ | $50.99_{\pm3.40}$ | $51.33_{\pm6.79}$ | $28.93_{\pm0.23}$ |
| Event-prior | $63.39_{\pm0.00}$ | $61.37_{\pm0.00}$ | $55.44_{\pm0.00}$ | $30.66_{\pm0.00}$ |
| Non-latent-attn | $88.49_{\pm0.74}$ | $68.50_{\pm2.07}$ | $83.59_{\pm1.67}$ | $29.90_{\pm0.63}$ |
| $+pri.$ | $89.83_{\pm0.88}$ | $69.99_{\pm1.35}$ | $83.58_{\pm6.23}$ | $30.38_{\pm1.49}$ |
| Non-latent-pert | $88.49_{\pm0.74}$ | $68.50_{\pm2.07}$ | $83.59_{\pm1.67}$ | $31.17_{\pm0.99}$ |
| $+pri.$ | $89.83_{\pm0.88}$ | $69.99_{\pm1.35}$ | $83.58_{\pm6.23}$ | $31.50_{\pm0.87}$ |
| **Single-article Models** | | | | |
| Two-player | $66.75_{\pm2.35}$ | $59.28_{\pm4.95}$ | $48.43_{\pm4.63}$ | $28.79_{\pm1.16}$ |
| $+pri.$ | $81.50_{\pm0.52}$ | $68.65_{\pm2.11}$ | $70.87_{\pm2.89}$ | $31.53_{\pm0.52}$ |
| Three-player | $66.87_{\pm2.32}$ | $60.15_{\pm2.36}$ | $48.74_{\pm3.55}$ | $29.72_{\pm2.30}$ |
| $+pri.$ | $81.06_{\pm0.86}$ | $65.60_{\pm0.55}$ | $70.57_{\pm2.51}$ | $30.70_{\pm2.68}$ |
| **Multi-article Models** | | | | |
| Two-player | $86.45_{\pm0.50}$ | $69.98_{\pm1.24}$ | $82.36_{\pm3.83}$ | $33.27_{\pm1.05}$ |
| $+res.$ | $85.68_{\pm0.32}$ | $68.01_{\pm2.93}$ | $82.38_{\pm3.28}$ | $33.54_{\pm0.91}$ |
| $+pri.$ | $91.03_{\pm0.72}$ | $71.27_{\pm1.14}$ | $84.31_{\pm5.58}$ | $33.32_{\pm0.74}$ |
| $+res.+pri.$ | **$91.58_{\pm0.25}$** | **$71.43_{\pm2.57}$** | **$89.16_{\pm3.04}$** | **$33.99_{\pm0.39}$** |
| Three-player | $85.48_{\pm1.47}$ | $65.08_{\pm1.57}$ | $83.10_{\pm1.82}$ | $33.20_{\pm3.25}$ |
| $+res.$ | $85.84_{\pm0.19}$ | $66.81_{\pm2.13}$ | $80.36_{\pm3.31}$ | $31.68_{\pm1.24}$ |
| $+pri.$ | $88.03_{\pm1.19}$ | $70.26_{\pm2.94}$ | $74.02_{\pm4.03}$ | $33.01_{\pm1.62}$ |
| $+res.+pri.$ | $88.02_{\pm1.45}$ | $69.39_{\pm1.01}$ | $80.92_{\pm3.57}$ | $31.46_{\pm1.75}$ |

Table 3: F1 scores (avg. of 5 runs) for ideology prediction and partisan event detection. $res.$: restrict models to prefer events selected by only one side; $pri.$: prior knowledge with pretrained event representations. Models that do cross-article comparison yield better performance on both tasks. Adding prior knowledge helps in almost all cases. Non-latent models have the same ideology prediction scores since they are the same model. **Best** results are in bold and second best are underlined.

take the majority vote among partisan events as article's ideology. Intuitively, this baseline utilizes the prior knowledge of event ideology to detect partisan events. (3) **Non-latent** is the best performing *multi-article* model in §3.3, which does not contain latent variables. Built upon this method, we create two variants for partisan event detection. The first is **attention**-based method, which is shown effective at finding input words that trigger the sentiment prediction (Wang et al., 2016). We use our trained model and consider the top $30\%$ of events with the largest attention weights (sum over all heads and positions) as partisan events. The second method is **perturbation**-based (Li et al., 2016), where we use the non-latent model and iteratively remove one event at a time and choose $30\%$ of events that lead to the largest output change as partisan events. We also report the performance of the multi-article model initialized from the pretrained event encoder in §4.3 for fair comparison with our latent models.

**Results.** Table 3 presents the results when $k = 30$, since $27.28\%$ of events in PEvent dataset are partisan. We explore the influence of $k$ in §6. Unsurprisingly, we observe that multi-article models

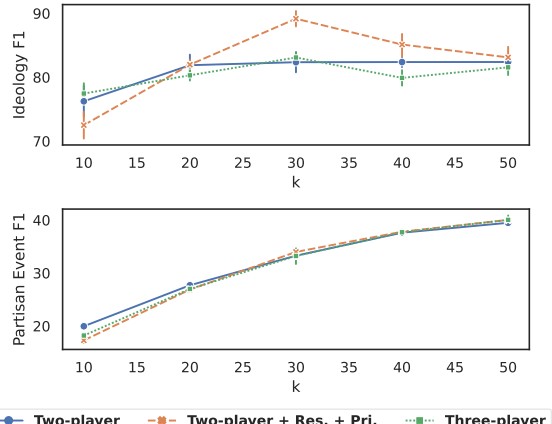

**Ideology Prediction/Partisan Event Detection F1**

Legend: Two-player, Two-player + Res. + Pri., Three-player

Figure 3: Ideology prediction and partisan event detection performance with different $k$ values (average of 5 runs). Error bars show the standard deviation. Performance variance is small for partisan event detection.

outperform single-article models on both tasks, emphasizing the importance of story-level context for cross-document event comparison.

On partisan event detection (last column of Table 3), *latent variable models outperform all baselines*, showing the effectiveness of training with article ideology labels. Note that the three-player models do not outperform the two-player models, indicating that the spurious correlation may not be a significant issue on PEvent, and partisan events annotated on PEvent, as standalone events, cannot be directly associated with specific ideological leaning. Moreover, restricting models from selecting common events improves partisan event detection for two-player models, which validates that *common events are less likely to be partisan*. Providing prior knowledge of event ideology further boosts on both tasks, especially for single-article models, illustrating the benefits of prior knowledge when the context is limited. Finally, combining the two improvements, the two-player model on the multi-article setup achieves the best performance. It is also important to point out that this model only uses 30% of events to predict ideology, but it still outperforms the model that sees full articles in the story, which suggests that a good modeling of events in the article could be more helpful than raw text representations when predicting ideology.

## 6 Further Analyses and Discussions

**Effect of Varying $k$.** We now explore the effect of $k$'s values. We experiment with three

multi-article models: base two-player, two-player with restriction and prior knowledge, and base three-player models. We train these models with $k = 10, 20, 30, 40, 50$ and plot the performance of partisan event detection and ideology prediction on PEvent in Fig. 3. For event detection, the model improves as $k$ increases, but the improvement is moderate when $k > 30$. For ideology prediction, the performance plateaus at $k = 20$ except for the two-player model with restriction and prior knowledge, which peaks at $k = 30$. This suggests that *only a subset of events reflect the article's ideology*, and it is enough to make predictions based on them.

**Error Analysis.** Table 4 and Table 10 in Appendix present predictions by the two-player model on the multi-article setup with one-sided restriction and prior knowledge for events. Two major types of errors are observed. First, the model struggles when an article attacks a statement from the opposite side with an implicit sentiment. For instance, "threw," "continue," and "had" in Table 4 are events or statements from the *right*, but the author reports them with an implicit negative sentiment (e.g., "not a thing!"), making the event *flatter to the left*. Future models need to (1) have an enhanced understanding of implicit sentiment along with the involving entities (Deng and Wiebe, 2015; Zhang et al., 2022), and (2) acquire knowledge of entity ideologies and their relations. Second, the model still frequently selects main events as partisan content, as shown by the "delivered" event in Table 10. For this example, it is because the main event should be included as necessary context for ideology prediction, i.e., the training objective. For other examples, some main events also carry sentiment towards ideological entities, thus indeed should be labeled as partisan events according to our definition. Future work should investigate whether the selective usage of partisan events are different when the main stories already support a certain ideology compared to when they disfavor the same ideology.

**Usage of the Latent Variable Model and Future Directions.** The latent variable model can be used as a *stance analyzer*, which would come in handy in practice as well, especially for generating *rationales* out-of-the-box in different settings. Firstly, being trained as an ideology predictor, it has the potential for future extensions to multi-modal ideology analysis, as suggested in Qiu et al. (2022). Secondly, working on articles on the same topics, it enables exploration of how different media outlets

**Title**: At the NRA Convention, People **Blame** Mass Shootings on Everything But Guns

The nation has been plunged into despair and mourning . . . in Houston, the National Rifle Association still **threw a party** . . . Two messages **emerged** from the assembled throngs and the doting politicians in attendance, just 300 miles from Uvalde: 1) People must **continue** to enjoy the right to acquire any damn firearm they choose, without meddling from the state; and 2) the massacre **had** absolutely nothing–not a thing!–to do with the untrammeled commerce in guns . . .

---

**Ideology label**: left      **Prediction**: left

Table 4: Article snippets of model predictions (multi-article two-player model with both improvements) and annotations. Colored spans denote events, with the **predicate** bolded. Blue: model predictions; red: human annotations; purple: annotations and predictions.

convey stances on shared topics by highlighting the inclusion or omission of partisan events. Lastly, our tool has the potential to complement entity-focused stance analysis. For instance, in the context of a left-leaning entity slashing a right-leaning entity, existing datasets only reveal a stance label between the two entities with no further rationale provided to the users (Deng and Wiebe, 2015; Zhang et al., 2022). In contrast, our analyzer can take an annotated stance as input and then identify the partisan event that influences readers' interpretation.

## 7 Conclusion

Partisan event selection is an important form of media bias which may exist in even the most apparently nonpartisan news, but which is especially hard to detect without extensive cross-article comparison. We first verify the existence of partisan event selection by inspecting the impact of partisan events on the performance of ideology prediction. We then jointly detect partisan events and predict article's ideology using latent variable models. Experiments show that detected partisan events reasonably align with human judgement, and our models using cross-article event context outperforms the counterpart that only uses single-article context on both two tasks. Our analysis also suggests future directions for identifying interactions among entities in an article and for resolving event coreference across articles.

**Title**: Biden calls for assault weapons ban, making gun manufacturers liable for shootings

President Biden on Thursday made an emotional appeal for ambitious new gun laws, including a ban on military-style rifles . . . On the other side of the aisle, Republicans **bristled** at Democrats' equating support for the Second Amendment with tolerating mass murder. "You think we don't have hearts," said Rep. Louis Gohmert, Texas Republican.

---

**Ideology label**: right      **Prediction**: left

Table 5: Article snippets where the extractor detects a right event, but the predictor predicts the article as left.

## Acknowledgments

This work is supported in part by the National Science Foundation under grant III-2127747 and by the Air Force Office of Scientific Research through grant FA9550-22-1-0099. We would like to thank the anonymous reviewers for their helpful comments and feedback.

## 8 Limitations

We investigate the impact of event selection on models' ideology prediction performance, to verify the existence of event selection in news media. The results, however, do not state a causal relation between media ideology and reported events.

We analyze the model output and discuss in details two major limitations of our latent variable models in §6. Apart from those errors, we also observe that events detected by the model as partisan may not align with the model's prediction of the article's ideology. In other words, the model could identify right-leaning events as partisan events while predicting the article as left-leaning (Table 5). Although the methods we adopt in this paper identify events that are indicative of ideology (Chen et al., 2018; Yu et al., 2019), they do not provide further justifications for how these events interact to reflect the ideology. For instance, the extractor could detect a right event and several left events that attack it. To further understand the event selection effect, future work may consider incorporating event-level ideology to model the interplay among events.

Although our models that include cross-article context can be extended to any number of articles without modification, they may be restricted by the GPU memory limit in practice. Particularly, the Transformer encoder that contextualizes all events

in a story requires computational resources to scale quadratically with the number of events, which is infeasible for stories that contain many articles. Future work may consider designing novel mechanisms to address this issue, e.g., by using special attention patterns based on the discourse role of each event in the article (van Dijk, 1988; Choubey et al., 2020).

Finally, due to the cost of manual labeling, we only evaluate our partisan event detection models on a dataset that covers two specific political issues. It remains to be seen whether methods introduced in this paper can be generalized to a broader range of issues. We call for the community's attention to design and evaluate partisan event detection models on more diverse topics.

## 9  Ethical Considerations

### 9.1  Dataset Collection and Usage

**Partisan Event Dataset Collection.** We conform with the terms of use of the source websites and the intellectual property and privacy rights of the original authors of the texts when collecting articles. We do not collect any sensitive information that can reveal original author's identity. We also consult Section 107[5] of the U.S. Copyright Act and ensure that our collection action fall under the fair use category.

**Datasets Usage.** Except the partisan event dataset collected in this work, we get access to the Basil dataset by direct download. For AllSides, we contact with the authors and obtain the data by agreeing that we will not further distribute it.

### 9.2  Usage in Application

**Intended Use.** The model developed in this work has the potential to assist the public to better understand and detect media bias in news articles. The experiments in §5 show that our model is able to identify partisan events on two controversial issues that moderately align with human judgement. The detected events can be presented to show different perspectives from both ends of the political spectrum, thus providing readers with a more complete view of political issues.

**Failure Modes.** Our model fails when it mistakenly predicts a non-partisan event as a partisan event, misses out the partisan events, or predicts the

wrong ideology for an article. They may cause misperception and misunderstanding of an event. For vulnerable populations (e.g., people who maybe not have the specific knowledge to make the right judgements), the harm could be amplified if they blindly trust the machine outputs.

**Biases.** The training dataset is roughly balanced in the number of left and right articles, so the model is not trained to encode bias. However, the dataset is relatively small and does not cover all possible political topics. Particularly, most of the news articles in the training set are related to U.S. politics, thus the model is not directly applicable to other areas in the world.

**Misuse Potential.** Users may mistakenly take the model outputs as ground truth. We recommend any usage of our model displaying an "use with caution" message to encourage users to cross-check the information from different sources and not blindly trust a single source.

---

[5] https://www.copyright.gov/title17/92chap1.html#107

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

# Appendix A    Implementation Details

For all experiments in this paper, our implementation is based on Pytorch (Paszke et al., 2019) and HuggingFace transformers (Wolf et al., 2020) library, and we preprocess all articles using Stanza (Qi et al., 2020). All experiments are conducted on 4 NVIDIA RTX A6000 GPUs.

# Appendix B    Event-based Ideology Prediction Models

## B.1    Event Extraction

We follow the scheme in TimeML which defines events as "situations that happen or occur" (Pustejovsky et al., 2003). We train an event extraction model on the MATRES data (Ning et al., 2018), as its event annotation is not limited to predefined event types, and thus is applicable to the open domain scenario. We use RoBERTa-large (Liu et al., 2019b) that predicts a binary label for each word, deciding whether the word is an event predicate or not. To provide surrounding context, we split articles into groups of 4 sentences and process 4 sentences together. We follow previous work on using TimeBank and AQUAINT sections in MATRES as training set and Platinum section as test set (Ning et al., 2019). Table 6 shows the hyperparameters for model architecture and training process. On the same train and test split, our model achieves an F1 score of 89.53, which is on par with the state-of-the-art performance of 90.5 F1 score (Zhang et al., 2021). As verbs and nouns account for $96.8\%$ of event predicates in MATRES dataset, we extract arguments 0 and 1 for verb and noun predicates using semantic role labeling tools (Shi and Lin, 2019; Gardner et al., 2018),[6] and we only keep predicates that match our event extraction results.

Multiple events can exist in one sentence with overlapping predicates and arguments. We hence remove the shorter event if there is an overlap, as we find that shorter events tend to be less informative. For example, it is easier to determine the partisanship of the event "the leak of a draft opinion would mark a stunning betrayal of the Court's process" than a shorter one on "the leak of a draft opinion." Therefore, we remove an event if its predicate is covered by another event's arguments.

| Hyperparameter | Value |
| --- | --- |
| number of epochs | 20 |
| patience | 4 |
| maximum learning rate | 3e-5 |
| learning rate scheduler | linear decay with warmup |
| warmup percentage | 6% |
| optimizer | AdamW (Loshchilov and Hutter, 2019) |
| weight decay | 5e-5 |
| # FFNN layer | 2 |
| hidden layer dimension in FFNN | 768 |
| dropout in FFNN | 0.1 |

Table 6: Hyperparameters used for the event extraction model.

## B.2    Contextualized Event Representation

Here we detail our model that uses cross-article context for ideology prediction. As described in §3.1, we first input the sentence that contains the event to a DistilRoBERTa model (Sanh et al., 2019) to get event representation $\mathbf{e}$. This representation is then passed to a Transformer encoder (Vaswani et al., 2017) with three embeddings to obtain contexualied event representation $\mathbf{c}$:

- **Article embedding** indicates the index of the article that contains the event, with one embedding per article index. The datasets we experiment with in this paper have at most 3 articles in each story. During training, we randomly shuffle the articles in each story.

- **Frequency embedding** informs the model whether the event appears in only one article, at least two but not all articles, or all articles in the story. We have one embedding per category. We find common events through lexical matching. Concretely, we use a dictionary that contains derivational morphology mappings (Wu and Yarowsky, 2020) to get the base form of the event predicate. We then construct a set of words for the predicate by including the synonyms for the base form and original form (Bird et al., 2009). Finally, two events are considered as the same if their predicate sets overlap and both of their ARG0 and ARG1 have a high word overlap (a threshold of $0.4$,[7] calculated by over-

---

[6] github.com/CogComp/SRL-English for nouns.

[7] We search threshold values from 0.2 to 0.5 by manually inspecting identified common events in 6 articles. A value of 0.4 can identify common events accurately while still allowing variations such as variants of mentions (e.g., president vs.

| Hyperparameter | Value |
|---|---|
| number of epochs | 5 |
| maximum learning rate | 5e-5 |
| learning rate scheduler | linear decay with warmup |
| warmup percentage | 6% |
| optimizer | AdamW |
| weight decay | 1e-4 |
| transformer hidden dimension | 768 |
| transformer # heads | 12 |
| # transformer layer | 4 |
| # FFNN layer | 2 |
| hidden layer dimension in FFNN | 768 |
| dropout in FFNN | 0.1 |

Table 7: Hyperparameters used for the event-based ideology prediction model.

lap coefficient, without stop words).

- **Position embedding** represents the relative position of the event in the article. We multiply the relative position of the event (a real number in $[0, 1]$) with a learnable embedding.

We further train a `[SEP]` token that separates the events from different articles. Finally, average representation of all events in an article is used to predict the article's ideology. Table 7 includes the hyperparameters of the model.

The entire model contains 106M parameters. On average, the training takes 25 minutes on a single NVIDIA RTX A6000 GPU.

## Appendix C  Partisan Event Annotation

**Data Collection.** We manually collect 25 stories, each with three articles from AllSides[8] that relate to the mass shooting in Texas and the overturn of *Roe v. Wade*. We extract events from each article and only keep the left and right article in each story.[9] We mask out the name of the media (e.g., "CNN" and "Fox News") in the article before annotation to avoid bias.

**Annotation Process.** We hire three college students proficient in English and familiar with discerning ideology under the context of U.S. political spectrum. We present each story, together with

---
president Biden).

[8]https://www.allsides.com/unbiased-balanced-news

[9]Each story on AllSides contains three articles from left, center, and right respectively. We only include the left and right articles in our dataset.

|  | AllSides | Basil | PEvent (ours) |
|---|---|---|---|
| # stories | 2,221 | 67 | 25 |
| # articles | 5,361 | 134 | 50 |
| # events detected per article | 66.82 | 48.71 | 60.70 |

Table 8: Statistics for AllSides training set, Basil (test only), and PartisanEvent (test only). AllSides **test** set contains 1,416 articles.

extracted events (predicate, ARG0, and ARG1) to annotators, without revealing the media source. The annotators are asked to first finish reading two articles on the same story but written by media of left and right leanings. They will then follow the steps below:

- Sort articles by their ideological position (left or right) in this story.

- Identify the main entities or pronouns in ARG0 and ARG1 of the event. The main entities can be the name of political groups/figures, bills/legislation, political movements or anything related to the topic of each article. If ARG0 and ARG1 are empty, identify the main entities or pronouns within the same sentence. Based on the context, try to resolve what event or entity each pronoun refers to.

- Estimate the sentiment toward each entity in the event. Sentiments can be reflected in words, quotations, and the relations between entities.

- Use entities and sentiments to decide whether the event is sided with the article's ideology. If it does, label it as a partisan event. Ex. Label an event as partisan in the left article, only if its "left" entity has a positive sentiment, or its "right" entity has negative sentiment. Also, it may be possible for events to be purely factual, which means there is no strong sentiment toward entities in events. For these kinds of events, try your best to estimate whether these events indirectly present any sentiment toward entities in the article.

Two annotators label all 50 articles, and a third annotator compares their annotations and resolve conflicts. We calculate inter-annotator agreement on all 50 articles and numbers can be found in §3.2.

**Partisan Events Distribution.** We further investigate the distribution of position of partisan events in the article. Fig. 4 shows the percentage of parti-

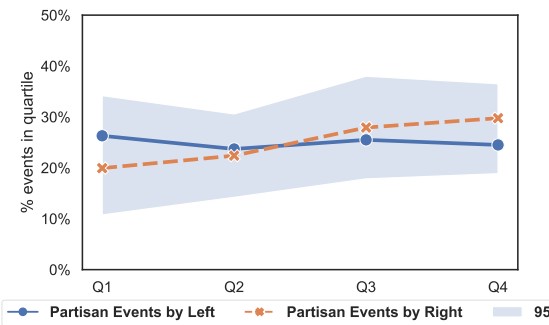

**Partisan Events Position Distribution**

Figure 4: Distribution of partisan events found in each quartile of an article, in terms of spatiality. Shaded area shows the 95% confidence interval.

san events that belong to each quartile of an article. As can be observed, right articles have more partisan events that appear in later parts of an article, whereas partisan events in left articles are evenly distributed in the article.

## Appendix D  Latent Variable Models

**Implementation Details.** For both extractor and predictors, we use the same model architecture as in §B.2 with hyperparameters listed in Table 7. For the three-player model, we follow the training process in Generative Adversarial Nets training (Goodfellow et al., 2014).

The two-player model contains 213M parameters, and the three-player model contains 320M parameters. On average, the training takes 50 minutes for the two-player model and 1.5 hours for the three-player model on a single NVIDIA RTX A6000 GPU.

**Pretrained Model for Event Representation.** We use the BIGNEWSALIGN dataset (Liu et al., 2022) to pretrain a model with prior event ideology knowledge. We remove stories in the dataset that contain duplicate articles and downsample articles in each story so that the number of left and right articles are balanced. Table 9 shows the statistics of the pretraining dataset. We then train a DistrilRoBERTa model that takes each event as input and predicts the event's ideology, where we use the article ideology as the event's ideology. We train this model on BIGNEWSALIGN for 2 epochs and use it to initialize our latent variable models.

## Appendix E  Additional Error Analysis

Table 10 in this section is supplementary to the Error Analysis section in §6. The model detects "de-

|       | # articles | # events  |
|-------|------------|-----------|
| Left  | 128, 481   | 6, 280, 732 |
| Right | 123, 380   | 4, 986, 165 |

Table 9: Statistics for the BIGNEWSALIGN pretraining dataset.

**Title**: "Enough": Biden **Exhorts** Congress To **Pass** Gun Control Laws

President Joe Biden **delivered** the second evening address of his presidency on Thursday night, almost **begging** Congress to **pass** gun control legislation . . . However, Biden **cited** former Supreme Court Justice Antonin Scalia-a conservative icon-who had declared that the Second Amendment was "not unlimited."

**Ideology label**: left    **Prediction**: left

Table 10: Article snippets of human annotations and model predictions (multi-article two-player model with both improvements). Highlighted spans denote events, with the **predicate** bolded. Blue: model predictions; purple: human annotations and predictions.

livered" as a partisan event, which is the main event in the story. The constraint introduced in §4.3 fails in this case because the other article describes this event differently (i.e., "Biden made an emotional appeal"), thus suggesting future research direction that leverages cross-document event coreference.