# OpenReview forum: "All Things Considered: Detecting Partisan Events from News Media with Cross-Article Comparison"
_EMNLP/2023/Conference — EMNLP 2023 Main_

### Official Review · Reviewer_9YKN · 2023-08-03

**Typos Grammar Style And Presentation Improvements:** The paper is very clearly and careful…
**Soundness:** 4

**Excitement:**

4: Strong: This paper deepens the understanding of some phenomenon or lowers the barriers to an existing research direction.

**Paper Topic And Main Contributions:**

This paper presents an approach to identifying the partisan stance of news articles through an analysis of partisan events.   The key observation is that events which are missing in an article (or located at the bottom of an article) can be highly predictive of stance given a notion of which events are more likely to appear in liberal or conservative sources.   The authors develop a series of models of increasing sophistication to exploit detections of events.

The main contributions are

(1) Introducing the use of partisan event detection for media ideology detection
(2) Comparing article level event analysis vs. contextualizing them through analysis of other articles on the topic from other sources.
(3) Preparing a modest annotated corpus on partisan events for evaluation and future studies.

**Questions For The Authors:**

Are there larger-scale studies which you will perform to use the stance predictor, or is the goal purely technological in a machine learning sense?

**Reasons To Accept:**


(1) The idea of models for stance identification using the presence and absence of events is appealing, and apparently understudied.
(2) Thorough evaluation of a range of different models for interpreting partisan events.
(3) Carefully and clearly written, a solid piece of work.

**Reasons To Reject:**

(1) The level of annotation (50 articles on only two topics) appears too small for a meaningful evaluation, although they leverage other existing data sets to get a meaninful evaluation

(2) Generally speaking, the performance of all the variants of the multiarticle models were pretty close, even though one model did win on all of the datasets.

(3) There was nothing "done" with the resulting stance analyzer other than validate its performance.   ideally one would build such an analyzer to conduct a study of news sources more than seek better F1 scores.

**Reproducibility:**

4: Could mostly reproduce the results, but there may be some variation because of sample variance or minor variations in their interpretation of the protocol or method.

**Reviewer Confidence:**

3: Pretty sure, but there's a chance I missed something. Although I have a good feel for this area in general, I did not carefully check the paper's details, e.g., the math, experimental design, or novelty.

---

> ### Author Rebuttal · Authors · 2023-08-29
>
> Thanks reviwer#9YKN for your constructive evaluation and comments.
>
> **Q1.** We acknowledge that a larger dataset with diverse topics would be beneficial. However, we also want to point out that the annotation is costly as it requires annotators with political domain background, and the annotation is done at the fine-grained event level. Compared to the existing informational bias dataset BASIL, our dataset contains **828 partisan events**, whereas theirs contains **1,249 informational bias spans**. However, all of our annotated partisan events are used for the **evaluation purpose**, whereas they use the informational spans for both **supervised training and evaluation**, resulting in a smaller test set of **62 sentences**.
>
> As you've noticed that we leverage other datasets for a meaningful evaluation, indeed we also report the ideology prediction performance on AllSides and BASIL (Table 4). Our hypothesis is that if the model can correctly detect partisan events, which by definition, will improve ideology prediction, which is validated by our experiments. We will leave the creation of a larger dataset for the future work.
>
> **Q2.** We acknowledge that the performance variance of multi-article models is small. However, our focus in this paper is not to develop a SOTA algorithm that outperforms existing methods, though our best model does end up with superior performance compared to baselines. Our goal is  to study the partisan events in news articles. We **verify their impacts** on the article's ideology, and we explore methods to detect partisan events **without supervised training** on large-scale datasets. The experiment results also verify that **cross-article event comparison** is important for partisan event detection and ideology prediction.
>
> **Q3 and additional questions.** Thanks for your interest in the use case of our model (i.e., stance analyzer). We would love to bring to your attention that the focus of this work is not only improving article-level ideology/stance prediction, but also providing rationales to the predicted stance so that readers would know **where the stance arises from**. For example, as presented in Fig 1, one may easily tell an article written by New York Post is right-leaning, while our analyzer further points readers to the specific right-leaning event (i.e., claims Biden’s statement as false) that emphasizes pro-gun rights.
>
> On top of that, our stance analyzer can be **employed in multiple settings**: (1) **Article-level ideology/stance prediction**, which is explored in this work (section 4). (2) **Topic-specific stances clustering** where we can investigate how different media express stances on the same topics by highlighting the inclusion or omission of partisan events. (3) **Entity-level stance analysis**: for instance, in the context of a left-leaning entity slashing a right-leaning entity, an existing dataset [1] only reveals a stance label between the two entities with no further rationale provided to the users. In contrast, our analyzer can take an annotated stance as input and then identify the partisan event that plays a role in influencing the reader's interpretation.
>
> [1] Generative Entity-to-Entity Stance Detection with Knowledge Graph Augmentation. (Zhang et al., EMNLP 2022)

---

### Official Review · Reviewer_MVtQ · 2023-08-04

**Soundness:** 4

**Excitement:**

4: Strong: This paper deepens the understanding of some phenomenon or lowers the barriers to an existing research direction.

**Paper Topic And Main Contributions:**

This paper addresses the gap in understanding media bias at the event selection level and introduces computational methods for detecting partisan events in news articles. By focusing on event selection rather than superficial biases, the study investigates how media ideology influences the choice of events to include in a news story. The authors define partisan events as selectively reported events that favor a media organization's partisans or are unfavorable to their opponents. The paper first shows the existence of partisan event selection by 1) article-level ideology prediction in single vs cross- article comparison setup and 2) removing partisan events from the articles for article-level ideology prediction. Then, the paper proposes the use of latent variables to represent whether an event is partisan or not, allowing for the development of computational tools without the need for manual annotation. This work sheds light on how news media content is produced and shaped, contributing to a deeper understanding of media bias's impact on society.


**Questions For The Authors:**

A. It is unclear why nonpartisan news content is mentioned in the following sentence: “providing new insights into high-level media bias even in seemingly nonpartisan news content.” Can you elaborate more on this point?


**Reasons To Accept:**

A1. Investigating media bias is an important societal challenge, and the study proposes new computational methods for studying media bias at the event selection level.

A2. The study formulates a new NLP task for partisan event detection in news articles and provides a new annotated dataset of 828 partisan events from 50 articles that focus on two political issues.

A3. It proposes latent variable models to detect partisan events, and these models outperform competitive baselines in both partisan event detection and ideology prediction tasks.

A4. The paper is very well written, and the methods and analyses sound strong.


**Reasons To Reject:**

R1. In the manuscript, the authors mention that the 'partisan coverage filtering' has a conceptual overlap with coverage bias (agenda setting) or framing bias. Unfortunately, the merit of this approach is not well elucidated. It would greatly enhance the clarity of the study if the authors could provide a more comprehensive explanation of how event-level selection bias contributes to the analysis of media bias.

R2. The authors' efforts in creating a new annotated dataset of partisan news events are appreciated. However, it is important to note that these events are derived from only two political issues, which may limit the generalizability of the study's findings.

R3. The study indirectly verifies the existence of partisan event selection bias by predicting the ideology of news articles. Nonetheless, it remains unclear whether those partisan events hold enough significance to sway audience beliefs and attitudes.

R4. The proposed model's performance for partisan event detection seems to show marginal improvement, with approximately a 5 percent point increase in the f1 score compared to the random model's score of 28.93.


**Reproducibility:**

4: Could mostly reproduce the results, but there may be some variation because of sample variance or minor variations in their interpretation of the protocol or method.

**Reviewer Confidence:**

4: Quite sure. I tried to check the important points carefully. It's unlikely, though conceivable, that I missed something that should affect my ratings.

---

> ### Author Rebuttal · Authors · 2023-08-29
>
> Thanks reviewer#MVtQ for your constructive feedback and assessment of our work.
>
> **R1.** Our current version might not have clearly articulated the difference among these concepts. We will polish them in our revised version as follows. Prior work on agenda setting is more on **topics-level**, e.g., whether abortion is something significant to talk about, and existing work on framing is more on **linguistic choices** by highlighting one particular aspect , e.g., pro-life frame v.s. Pro-choice frame. In contrast,  our work on partisan coverage filtering is on the subtle way by which the media sway their readers through **conscious inclusion or omission of some certain events or subevents**. Eventually, this fundamentally affects how readers will interpret the stories through salience estimation and causal relation understanding.
>
> As for the significance of event-level selection bias to the media bias analysis, we will elaborate it more in the revision. Below is our brief response:
>
> We would like to first stress the **difficulty of locating a reported biased event**. This is mainly due to the fact that an event evinces partisanship only under certain circumstances even if an event itself might not be biased. Detecting such a partisanship is non-trivial since partisan events are meticulously and consciously selected by the writer in a subtle way to make the news content maintain apparent objectivity (see our response to R4). Our work is the first that advances effective partisan event extraction out of a set out seemingly neutral events, which would in turn allow people to spot out where the media bias emerges.
>
> Further, traditional techniques tackling media bias challenges are only able to predict article or media-level ideological slant. In contrast, our model enhances models’ **reliability and interpretability** by explicitly informing the audiences of potentially biased events (i.e., detected partisan events).
>
> Another merit of our approach is that it **does not rely on annotated partisan event labels**, which makes it easier to adapt to unseen topics since article-level ideology labels are cheaper to get.
>
> **R2.** We admit that the evaluation results might only adequately reveal our models’ performances on two representative topics in 2022. However, it is worth noting that the two topics in our annotated PEvent are only used for evaluation, while our models are trained on AllSides, which covers a disjoint set of topics. The improved performance on the cross-topic setting **demonstrates the generalizability of our model**. Therefore, we conjecture that models would follow a similar trend when performing on other common yet controversial topics. We will leave a larger-scale evaluation to the future study.
>
> **R3.** As validated in Broockman and Kalla (2022), researchers in the social science community have unveiled the existence of partisan event selection bias, which has a persuasion effect to sway audience beliefs and attitudes. Our work, motivated by their survey-based empirical finding, **develops computational tools to verify the existence** in the following two aspects:
>
> (1) We inspect the impact of partisan events on the performance of ideology prediction to indirectly validate the existence (section 3). (2) Due to the absence of annotated partisan events in the literature, we curate PEvent by directly annotating partisan events. The PEvent dataset illuminates that **partisan events are ubiquitous in news and account for around 30% of reported events** (section 4). We hope that our tools and findings can be used in future studies of assessing the belief change and attitude change by the public at scale.
>
> **R4.** We acknowledge that the best model achieves incremental improvement. However, considering the subjectivity and difficulty of the task where human annotators can only achieve 52.83 F1 score (using one annotator's labels as ground truth and another's as prediction), the improvement (33.99 vs 28.93) is non-trivial.
>
> **Additional Questions.** In journalism, journalistic objectivity is a foundational principle that instructs journalists to report news content factually and non-partisanly. However, the actual news reporting is inevitably influenced by ideological slant of media outlets or article writers. In order to maintain a balance between objectivity principle and ideological slant, writers selectively include or omit some certain events (i.e., partisan events) without the usage of partisan words, which in turn still sways people’s interpretation. However, readers might falsely deem such an article non-partisan, though, it is not as we and other works argue, because of the absence of partisan words, so we call articles of this kind **seemingly non-partisan**.

---

### Official Review · Reviewer_bwkA · 2023-08-06

**Soundness:** 3

**Excitement:**

4: Strong: This paper deepens the understanding of some phenomenon or lowers the barriers to an existing research direction.

**Paper Topic And Main Contributions:**

The paper studies the problem of detecting patisan events, i.e. selectively reported events that are flattering to a media organization’s copartisans or unfavorable to opponents. The paper defines patisan events detection as a NLP task, where given an article, a model is expected to extract all events in the form of predicate-argument triples, and identify the set of partisan events. To facillitate evaluation on the task, the paper introduces a benchmark dataset. The paper offers a latent variable model as baseline for the task, and conduct analysis to demonstrate the effectiveness and weakness of the model.

**Questions For The Authors:**

See reasons to reject above -- Happy to hear what the authors think and any clarifications that they might have. I will adjust my ratings accordingly.

**Reasons To Accept:**

The parisan event detection task could be an interesting reserach topic for the community,
The proposed dataset could facilitate future research and evaluation on the task.
The drop rate analysis for ideology prediction after removing partisan events is interesting, and supports the motivation of the study.
The paper uses a latent variable model to demonstrate the benefit of using partisan event detection to improve ideology prediction.

**Reasons To Reject:**

In the paper's current form, many of the experimental settings are underspecified, and could be greatly improved by another round of revision. Specifically, here are some comments and suggestions --

1. Personally I'm having a hard time understanding the task format of event detection. During evaluation, are the models expected to extract the events, or if the gold events are given as input for each document? Given how the authors describe the "evaluation metrics", I'm guessing the gold events are given? However, section 4.1 "task overview" mentions "we extract events" from document, which seem to suggest that event extraction is a part of the task??

2. Along the previous point, It's probably worth having a separate section for task definition, perhaps at the beginning of section 3.  What Section 4.1 "Task Overview" offers is a description of the inputs/outputs of latent variable models, but the precise definition of task or the format in which the task is evaluated is still unclear.

3. Ideally, please include the % of "partisan" events in train/test/dev split, as understanding whether the model is trained in similar label distritbution is important for readers to understand your baseline performance in context. The random baseline is getting 28.93%, which seems to suggest there are 28.93% in test set, while the paper mentions "27.28% of events in PEvent dataset are partisan".  So I would've guessed that all splits have similar distribution. (A reader of your paper shouldn't really be guessing for such things!)

With the evaluation + experimental settings clarified, I would tend to rate it as a solid paper. But in its current form, presentation needs to be improved.

**Reproducibility:**

4: Could mostly reproduce the results, but there may be some variation because of sample variance or minor variations in their interpretation of the protocol or method.

**Reviewer Confidence:**

4: Quite sure. I tried to check the important points carefully. It's unlikely, though conceivable, that I missed something that should affect my ratings.

**Typos Grammar Style And Presentation Improvements:**

Here are a few structral improvements that I would suggest for the paper --
1. Have a "task definition" section and describe the evaluation format of the task without considering the solution. This can be in the beginning of section 3.
2. Dedicate more space for result analysis, so that people understand (1) What baseline models can do currently, and (2) If people are going to work on your problem, where can potential improvements come from.
3. Maybe revise figure 1, so that the task setting is clear from the get-go. In its current form, it looks like the task requires the model to extract the event as well.  (I still have no clue what the setting is up to now.)

---

> ### Author Rebuttal · Authors · 2023-08-29
>
> We would like to first thank reviewer#bwkA for your time and thoughtful input. Please find our responses to your questions below.
>
> **Q1.** The event extraction is not part of our partisan event detection task (section 4). Concretely, we assume the inputs to our ideology prediction model in section 3 and latent variable model in section 4 are **events** instead of articles. These events are extracted by an off-the-shelf event extraction model (Appendix B.1) in the format of triplets of  <ARG0, predicate, ARG1>. The event representations are based on contextual representations derived from the articles.
>
> To annotate partisan events in the PEvent dataset, we use the same event extraction model to extract events from an article in the first place. Annotators are then asked to label partisan events **among the extracted events**. We will make sure this is clearly stated in the revision
>
>
> **Q2.** Thanks for the suggestion. We will add a task definition section to clarify task input/output format at the beginning of section 3.
>
>
> **Q3.** We would like to point out that we only annotate partisan events on the PEvent dataset, which is only used for the **evaluation purpose**. Our training/validation sets **do not contain labeled partisan events**. Instead, we train the latent variable models using the article-level ideology labels, and the identified partisan events are what the model “believes” to be the most indicative of the article-level ideology.
>
> The partisan event detection F1 score for the random baseline reported in the paper (28.93%) is the average performance of 5 runs. The expected (i.e., average of infinite numbers of runs) F1 score for the random baseline should be 28.58% (precision of 27.28% and recall of 30.00%).
>
>
> **Presentation improvements.** Thanks for the suggestions. We will incorporate them in the revision.
>
>
> We hope we have addressed all the concerns you raised about our work and hope you can reconsider your evaluation. If there are any other issues, please let us know by updating your review, and we would be happy to resolve them in the revision.

---

### Meta-Review · Area_Chair_CUkd · 2023-09-17

**Recommendation:** 3

**Metareview:**

Quality: The paper presents an interesting and relevant task. The authors have conducted experiments to back up their claims, however the experiments are lacking details in the current version of the work. Clarifications that were provided in the rebuttals should be added in to the final version of the paper.

Clarity: The paper is grammatically sound and well written. However, some reviewers suggested better organization to make the task definition more clear and easier to follow.

Originality: The paper proposes partisan event detection in news articles, which can be seen as a novel task since most works focus on sentence or document level ideology classification. Additionally, a joint latent model is introduced for solving this task. All the reviewers liked the idea and motivation of the work.

Significance: Computational political science researchers in NLP will be interested in this work. Detection of media bias is also incredibly important in today's society. However, the lack of experimental details, low inter-annotator agreement on a small dataset (~800 events, 2 political issues), marginal improvement over a minimal baseline, and limited discussion of the application of this work towards understanding framing and bias in the media are considerable drawbacks.

Pros: (1) Proposing an important societal task of partisan event detection, which can be seen as novel compared to typical sentence-level (or higher) ideology detection.
(2) A latent model for event detection, which can also be leveraged in an unsupervised setting.
(3) A new dataset of 50 articles on 2 political issues, with annotation provided for 828 events.

Cons: (1) Underspecified experimental settings which led to reviewer confusion about the task and evaluation. While this was clarified to the reviewers' (bwkA & MVtQ) satisfaction in rebuttal, this missing information is critical and should be included in the final version of the paper.
(2) The weak performance of the model and lack of further application is a concern to the reviewers. The random baseline performance is quite low and the latent model has marginal improvement.
(3) The dataset is very small. Of 50 articles, 3035 events were extracted. Of these, 828 events were labeled and used in this work. Also, the reported inter-annotator agreement of 0.43 (and 52.83 in rebuttal) is too low.
(4) Lack of discussion on the link between ideology, agenda setting, and framing bias; specifically more explanation is needed for how event selection bias contributes to the analysis of media bias. Inclusion of rebuttal points to Reviewer MVtQ could strengthen this area. (5) In the author's rebuttals to Reviewer MVtQ & 9YKN: Please note that the Broockman and Kalla term partisan coverage filtering is essentially framing (here, at the event level). Also, the proposed task of topic-specific stance clustering is again framing of events. The meta-reviewer recommends further reading into the interplay of political ideology, framing, and media bias to strengthen the contributions of this work.

Overall, the reviewers really like the idea and motivation of the proposed task. Inclusion of the details that were provided in the author's rebuttals would greatly strengthen the paper.

---

### Decision · Program_Chairs · 2023-10-07

**Decision:**

Accept-Main

**Comment:**

Quality: The paper presents an interesting and relevant task. The authors have conducted experiments to back up their claims, however the experiments are lacking details in the current version of the work. Clarifications that were provided in the rebuttals should be added in to the final version of the paper.

Clarity: The paper is grammatically sound and well written. However, some reviewers suggested better organization to make the task definition more clear and easier to follow.

Originality: The paper proposes partisan event detection in news articles, which can be seen as a novel task since most works focus on sentence or document level ideology classification. Additionally, a joint latent model is introduced for solving this task. All the reviewers liked the idea and motivation of the work.

Significance: Computational political science researchers in NLP will be interested in this work. Detection of media bias is also incredibly important in today's society. However, the lack of experimental details, low inter-annotator agreement on a small dataset (~800 events, 2 political issues), marginal improvement over a minimal baseline, and limited discussion of the application of this work towards understanding framing and bias in the media are considerable drawbacks.

Pros: (1) Proposing an important societal task of partisan event detection, which can be seen as novel compared to typical sentence-level (or higher) ideology detection.
(2) A latent model for event detection, which can also be leveraged in an unsupervised setting.
(3) A new dataset of 50 articles on 2 political issues, with annotation provided for 828 events.

Cons: (1) Underspecified experimental settings which led to reviewer confusion about the task and evaluation. While this was clarified to the reviewers' (bwkA & MVtQ) satisfaction in rebuttal, this missing information is critical and should be included in the final version of the paper.
(2) The weak performance of the model and lack of further application is a concern to the reviewers. The random baseline performance is quite low and the latent model has marginal improvement.
(3) The dataset is very small. Of 50 articles, 3035 events were extracted. Of these, 828 events were labeled and used in this work. Also, the reported inter-annotator agreement of 0.43 (and 52.83 in rebuttal) is too low.
(4) Lack of discussion on the link between ideology, agenda setting, and framing bias; specifically more explanation is needed for how event selection bias contributes to the analysis of media bias. Inclusion of rebuttal points to Reviewer MVtQ could strengthen this area. (5) In the author's rebuttals to Reviewer MVtQ & 9YKN: Please note that the Broockman and Kalla term partisan coverage filtering is essentially framing (here, at the event level). Also, the proposed task of topic-specific stance clustering is again framing of events. The meta-reviewer recommends further reading into the interplay of political ideology, framing, and media bias to strengthen the contributions of this work.

Overall, the reviewers really like the idea and motivation of the proposed task. Inclusion of the details that were provided in the author's rebuttals would greatly strengthen the paper.